# Reproducibility Report: Contrastive Learning of Socially-aware Motion Representations

## Reproducibility Summary

The following paper is a reproducibility report for "Social NCE: Contrastive Learning of Socially-aware Motion Representations" [1] published in ICCV 2021 as part of the ML Reproducibility Challenge 2021. The original code was made available by the author [1]. We attempted to verify the results claimed by the authors and reimplemented their code in PyTorch Lightning.

**Scope of Reproducibility**

The central claim of the paper is that the consideration of negative (collision) cases in trajectory prediction models through a socially contrastive loss function Social-NCE will improve the robustness of the models. We verify their claim on various models, with special focus on improvements in the human trajectory prediction models Social-STGCNN and Trajectron++ and on robot navigation through an imitation learning model.

**Methodology**

We used the codebase made publicly available by the authors for our work. We trained the models used in the paper from scratch and reimplemented the code in PyTorch Lightning. We evaluated both, and compared them with the results in the original paper. Further, we attempted additional experiments to find suitable hyperparameters in the Trajectron++ and Social-STGCNN models.

**Results**

We were able to reproduce majority of the results claimed in the paper except the Social-LSTM and Directional-LSTM models due to lack of time, and got a maximum of 2% deviation from that of the original paper.

**What was easy**

The publicly available codebases were well documented and easy to follow. The authors have also mentioned sources for the processed datasets that they have used. The simulation data generation code for the imitation learning model was also shared.

**What was difficult**

The proposed contrastive loss was implemented on different trajectory prediction models, the understanding of which was required to reimplement the code from PyTorch to PyTorch Lightning. Experiments on the entire ETH and UCY dataset on restricted computational resources took a considerable amount of time and we had to restrict our ablation study to one model.

**Communication with original authors**

We contacted the authors with some queries on their implementation and on the importance of some hyperparameters. They replied promptly and their input was pivotal while conducting experiments.

---

[1]https://github.com/vita-epfl/social-nce

# 1 Introduction

Humans tend to develop a strong intuition towards predicting future motions of other people, while navigating in crowded spaces. This is essential for carrying out daily tasks without any discomfort and to maintain a safe distance from others while moving around. However, building neural models that can replicate similar nature of accurate predictions is often challenging, even with a large training set.

Multi-agent problems such as trajectory forecasting and robot navigation, require the model to learn socially aware motion representations. Previously, several papers have proposed neural network based models to achieve these tasks. However, these models still fail to generalize well with different scenarios, often outputting colliding trajectories. The original authors aim to tackle this issue by feeding explicit negative examples into the network, while teaching the model to differentiate between the two using a newly proposed Social Contrastive Loss.

We exhaustively carry out all the experiments done in the paper and verify all claims and tables. We then review the results and present an assessment. We further ported the code to the PyTorch Lightning framework. This allowed us to train the code flexibly over different platforms and automate the optimization process. We also expect this to help in future implementation or reproduction of the codebase. Then we proceed to present a few ablations in the original code, especially hyperparameter tuning.

# 2 Scope of reproducibility

Existing work on multi-agent trajectory prediction problems sometimes output colliding trajectories which makes them unsuitable for deployment. The authors claim that this is due to the bias in existing datasets which only consist of safe trajectories and no collision scenarios, giving the models no negative cases to train on. The original paper proposes a modified contrastive loss (Social-NCE) which incorporates ground truth knowledge to generate negative cases to reduce collision rates on several benchmarks. The details of this loss have been discussed later (in Methodology section) in the report. The key claims that we aim to verify in our reproducibility report are:

1. Addition of the Social-NCE loss in human trajectory forecasting models significantly decreases collision rate while maintaining similar final displacement error.

2. Addition of the Social-NCE loss in imitation learning models for robot navigation in crowded environments significantly decreases the collision rate.

3. Addition of the Social-NCE loss in reinforcement learning models increases sample efficiency, and they obtain a collision-free policy quickly.

# 3 Methodology

The authors have a detailed public repository[2] on the addition of Social-NCE on Trajectron++ [2], Social-STGCNN [3], models for human trajectory prediction and on an existing imitation learning model [4] for robot navigation. Further, we contacted the authors and they gave us their implementation of Social-NCE in reinforcement learning using Rainbow DQN [5] as the baseline. We reproduced the findings of the paper based on these repositories. We focused primarily on the human trajectory prediction models Social-STGCNN and Trajectron++ and attempted ablations on Social-NCE hyperparameters in the Trajectron++ model to improve its performance. Lastly we ported the codebase for Trajectron++, Social-STGCNN and the imitation learning model to PyTorch Lightning [6].

---

[2]https://github.com/YuejiangLIU/social-nce-trajectron-plus-plus

### 3.1 Social-NCE Loss and Negative Data Augmentation

Consider M agents with index $i \in \{1...M\}$, the state of agent $i$ at time $t$ is given by $s_t^i = (x_t^i, y_t^i)$ which are its position coordinates. State of all agents combined is given by $s_t = \{s_t^1, s_t^2....s_t^M\}$. Given $s_{1:t}$ the model predicts $s_{t+1:T}$.

**Encoder** $f(\cdot)$ **:**
Gives vector encoding( $h_t^i$) for agent $i$ at time $t$ given state of all agents till time t and index of agent:

$$h_t^i = f(s_{1:t}, i) \tag{1}$$

Encoder has two sub-modules: sequential $f_s(.)$ and interaction $f_i(.)$ modules to make encoding of one agent dependent on the state of other agents.

**Decoder** $g(\cdot)$ **:**
Returns predicted state from vector encoding

$$s_{t+1:T}^i = g(h_t^i) \tag{2}$$

#### 3.1.1 Social-NCE loss

**Embedding Models**

- **Query:** Projection head that embeds the vector encoding of the agent $i$ till time $t$

$$q = \psi(h_t^i) \tag{3}$$

- **Key:** Encoder that embeds the future state of agent i at time $t + \delta t$ where $\delta t$ is the sampling horizon in a given range

$$k = \phi(s_{t+\delta t}^i, \delta t) \tag{4}$$

Both the query and key are 2-layer MLPs which return 8-dimensional encoded vectors.

**Loss**
The InfoNCE Loss [7] is given by:

$$L_{NCE} = -log \frac{exp(sim(q, k^+)/\tau)}{\sum_{n=0}^{N} exp(sim(q, k_n)/\tau)} \tag{5}$$

In standard InfoNCE loss the similarity function sim(q, k) is the cosine similarity between the two vectors. In the Social-NCE variation this similarity function has been modified to the dot product of the two embedded vectors returned from the encoders. The Social-NCE Loss is given by:

$$L_{Social-NCE} = -log \frac{exp((\psi(h_t^i) \cdot \phi(s_{t+\delta t}^{i,+}, \delta t)/\tau)}{\sum_{\delta t \in \Lambda} \sum_{n=0}^{N} exp((\psi(h_t^i) \cdot \phi(s_{t+\delta t}^{i,n}, \delta t)/\tau)} \tag{6}$$

The three encoders $f(\cdot)$, $\psi(\cdot)$ and $\phi(\cdot)$ are jointly trained such that the query is encoded closer to the positive key and further from the negative keys. The keys are made through data augmentation as discussed next. The final loss for a specific model would be given by the weighted sum of the model task loss and the Social-NCE loss.

#### 3.1.2 Data Augmentation

Negative samples: The state of the agent i at time $t + \delta t$ cannot be same as the state of any of the other agents at time $t + \delta t$, so the states of the $M - 1$ elements other than the agent i can be used as negative keys for it.
For each agent $j \in \{1, ...M\} - \{i\}$, 8 points are taken uniformly from a circle with radius of minimum distance of comfort around the agent j as negative keys for agent $i$

$$s_{t+\delta t}^{i,n-} = s_{t+\delta t}^j + \Delta s_p + \epsilon \tag{7}$$

97  $\Delta s_p = (\rho cos\theta_p, \rho sin\theta_p)$, $\rho$ being minimum distance of comfort and $\theta_p = 0.25p\pi, p \in \{0, 1, ..., 7\}$

98  $\epsilon$ is a normally distributed added noise

99  Each agent $i$ thus has $8(M - 1)$ negative keys.

100  Positive samples: Single positive key is taken from state of agent $i$ at time $t + \delta t$ after adding normally distributed noise $\epsilon$

101

$$s_{t+\delta t}^{i,+} = s_{t+\delta t}^i + \epsilon \tag{8}$$

102  The data augmentation is made clearer by the following diagram given by the authors [1]. For an agent $i$ (in blue) the areas of Collision and Discomfort as shown are used as negative samples.

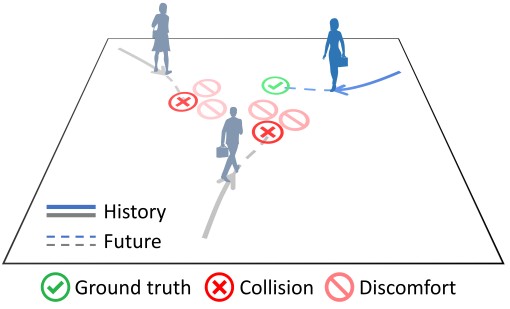

Figure 1: Social Negative Augmentation

103

## 3.2  Datasets

105  The human trajectory prediction models were run on a processed version of ETH and UCY datasets. The original
106  dataset is a collection of 5 video segments of pedestrian trajectories from which the states of each agent per frame
107  id had been stored and the dataset had been pre-divided into train, test and validation sets to maintain uniformity in
108  accuracy comparison. The processed ETH and UCY datasets are are available in the repository linked[3].
109  The imitation and reinforcement learning models used pedestrian data from an open-source simulator based on OpenAI
110  gym library [4]. The dataset[5] consisted of 5000 simulated situations in which the position of 5 random agents are stored
111  for each time step. A validation split of 0.3 was taken.

## 3.3  Hyperparameters

113  Apart from the hyperparameters required for regular network training, the Social-NCE included three additional
114  hyperparameters, specific to the model. These were: the temperature hyperparameter $\tau$, the sampling horizon $\delta t$ and the
115  contrastive weight $\lambda$. In the original paper, there values were set by default. We improvised upon previous work by
116  performing a thorough random search for these hyperparameters using WandB [8]. We further do a sensitivity analysis,
117  and check whether hyperparameter tuning offers any significant benefit. The details of the search can be summarised as
118  follows:

Table 1: Search on model hyperparameters

| Hyperparameter | Original Value | Method of Search | Range of Search |
|---|---|---|---|
| Temperature $\tau$ | 0.1 | Random | 0.1 - 0.5 |
| Sampling Horizon $\delta t$ | 4 | Grid | 1, 2, 3, 4, 5 |
| Contrastive Weight $\lambda$ | 2 | Random | 0 - 50 |

119  Details on the loss hyperparameters:

---

[3]https://github.com/StanfordASL/Trajectron-plus-plus/tree/master/experiments/pedestrians/raw
[4]https://github.com/vita-epfl/CrowdNav/blob/master/crowd$_s$im$/README.md$
[5]https://drive.google.com/uc?id=1D2guAxD$_EgrKnJFMcLSBkf$10$SOagz$0mr

- Temperature($\tau$): Part of the Social-NCE loss which controls the weight of the penalty and reward for negative and positive samples respectively.

- Sampling Horizon($\delta t$): The future time step till which the negative samples are considered for data augmentation

- Contrastive Weight($\lambda$): The weight between the main loss of the model and the Social-NCE Loss

A similar search was performed separately for the hyperparameters pertaining to data augmentation, with the default values for the hyperparameters discussed in the previous section. These hyperparameters were: Minimum Separation, Maximum Separation and the weight between maximum separation and noise. The details of this search can be summarised as follows:

Table 2: Search on hyperparameters of data augmentation

| Hyperparameter | Original Value | Method of Search | Range of Search |
|---|---|---|---|
| Minimum Separation | 0.2 | Random | 0.1 - 0.5 |
| Maximum Separation | 2.5 | Random | 2.2 - 2.8 |
| Weight between maximum separation and noise | 0.2 | Random | 0 - 0.5 |

Details on the augmentation hyperparmeters:

- Minimum Separation: Minimum admissible value of $\rho$ in negative augmentation which is the minimum comfortable distance between two agents

- Maximum Separation: Maximum admissible value of $\rho$ in negative augmentation which is the maximum distance after which agents can pass each other with collision.

- Weight between maximum separation and noise: The weight between the added normal noise and the position of the augmented sample.

### 3.4 Experimental setup and code

The encoder models were trained with Adam Optimizer. For the training of the Trajectron++, Social-STGCNN and imitation learning models 300, 500 and 200 epochs were used respectively. There were two runs of the reinforcement learning model on 2000 and 5000 episodes respectively. As mentioned in the original paper, the models were evaluated on the following metrics:

- Final displacement error (FDE): the Euclidean distance between the predicted output and the ground truth at the last time step.

- Collision rate (COL): the percentage of test cases where the predicted trajectories of agents run into collisions.

A lower FDE is preferred, however the current reproduction mainly aims to see the decrease in collision rate. The code was also integrated with WandB to conduct further experiments. This process involved constructing a config dictionary, which included the list of all possible hyperparameters and the values it could potentially take. The main function was modified with WandB initialisation and the logging function to log the value of the Loss after training is complete. The function was then passed to the WandB agent to carry out sweeps. The code can be found in this link [6].

### 3.5 Computational requirements

The training code for Trajectron++ and Social-STGCNN was run on Kaggle with GPU (Tesla P100-PCIE-16GB) and CPU (13GB RAM + 2-core of Intel Xeon).The average training runtimes are listed in the tables below. It can be seen clearly that porting to lightning has not caused any increase in training time.

---

[6]https://anonymous.4open.science/r/social-nce-stgcnn-62D5/README.md

Table 3: Training Runtimes

| Codebase | Original Codebase Training Time | Ported Codebase Training Time |
|---|---|---|
| Trajectron++ | 4hr 28mins | 4hrs 24mins |
| Social-STGCNN | 8hr 32mins | 8hr 30mins |
| Imitation Learning | 53min | 50mins |

## 4 Results

The following experiments support the claims made by the authors. We compared the results from training the model from scratch(Reproduced) and reimplementing the model in PyTorch Lightning (Ported Code) with the results given by the authors (Original Paper).

### 4.1 Results reproducing original paper

A comparison of the FDE (Final Displacement Error) and COL (Collision Rate) for the addition of Social-NCE to Trajectron++ and Social-STGCNN models in original paper, reproduced and ported code.

Table 4: Social-STGCNN

| Dataset | Ported Code | | Reproduced | | Original Paper | |
|---|---|---|---|---|---|---|
| | FDE | COL | FDE | COL | FDE | COL |
| ETH | 1.442 | **0.53** | 1.249 | 1.11 | **1.224** | 0.61 |
| Hotel | **0.598** | 3.49 | 0.681 | **3.25** | 0.678 | 3.35 |
| Univ | **0.856** | **6.39** | 0.878 | 6.44 | 0.879 | 6.44 |
| Zara1 | **0.492** | 1.29 | 0.515 | **1.02** | 0.515 | **1.02** |
| Zara2 | **0.453** | 3.58 | 0.481 | **3.26** | 0.482 | 3.37 |
| Average | 0.768 | 3.05 | 0.761 | 3.02 | **0.756** | **2.96** |

Table 5: Trajectron++

| Dataset | Ported Code | | Reproduced | | Original Paper | |
|---|---|---|---|---|---|---|
| | FDE | COL | FDE | COL | FDE | COL |
| ETH | **0.632** | 0.00 | 0.791 | 0.00 | 0.791 | 0.00 |
| Hotel | 0.193 | **0.29** | **0.163** | 0.32 | 0.177 | 0.38 |
| Univ | **0.426** | **2.95** | 0.442 | 3.29 | 0.435 | 3.08 |
| Zara1 | 0.439 | 0.18 | 0.338 | **0.14** | **0.330** | 0.18 |
| Zara2 | 0.452 | 0.95 | 0.281 | 1.02 | **0.255** | 0.99 |
| Average | 0.428 | **0.88** | 0.403 | 0.95 | **0.398** | 0.93 |

A comparison of the collision rate and time taken for the robot to reach destination for the addition of Social-NCE in the imitation learning model in original paper, reproduced and ported code.

Table 6: Imitation Learning

| Code | Time(s) | Collision(%) |
|---|---|---|
| Original | 10.33 | 3.40 |
| Reproduced | 10.49 | **3.36** |
| Ported | **10.28** | 3.45 |

A table of reward vs number of episodes trained for the implementation of Social-NCE on the reinforcement learning model.

Table 7: Reinforcement Learning

| Episodes | 0 | 1000 | 2000 | 3000 | 4000 | 5000 |
|---|---|---|---|---|---|---|
| Reward | -0.10 | 0.42 | 0.61 | 0.63 | 0.64 | 0.64 |

## 4.2 Hyperparameter tuning

The performance of any model, critically depends on the choice of hyperparameters. In the original paper, the values of these hyperparameters were set by default. We identified critical hyperparameters, specific to the Social-NCE, and conducted a thorough hyperparameter search, to find out the best possible combination of hyperparameters. Due to lack of time, this was done only on the Social STGCNN model and trained over the ETH dataset. The following table summarises the results of the search:

Table 8: Hyperparameter Search

| Hyperparameter | Original Value | Best Value |
|---|---|---|
| Temperature $\tau$ | 0.1 | 0.1412 |
| Sampling Horizon $\delta t$ | 4 | 1 |
| Contrastive Weight $\lambda$ | 2 | 16 |

A similar search was performed separately for the hyperparameters pertaining to Data Augmentation:

Table 9: Hyperparameter Search

| Hyperparameter | Default Value | Best Value |
|---|---|---|
| Minimum Separation | 0.2 | 0.22 |
| Maximum Separation | 2.5 | 3.1 |
| Weight between maximum separation and noise | 0.2 | 0.24 |

The FDE and collision rate after training the Social-STGCNN model for 400 epochs on the original (Original Parameters) and tuned hyperparameters(Tuned Parameters) are:

Table 10: Metrics on Original and Tuned Hypeparameters

| Model | FDE | COL |
|---|---|---|
| Tuned Parameters | 0.674 | 3.45 |
| Original Parameters | 0.678 | 3.54 |

## 5 Discussion

Our results support the authors' claim that modelling of social knowledge through the addition of negative test cases reduce the collision rate of trajectory prediction models. In both training from scratch in the original code and in the ported code, the results have remained consistent with that of the original paper.

1. In human trajectory forecasting, the addition of Social-NCE to the models Trajectron++ and social-STGCNN showed a 35.7% and 35.1% decrease in collision rate on average respectively(Table 4 and Table 5) in our reproduced results. The Final Displacement Error(FDE) showed deviation of less than 1%, showing that addition of Social-NCE adds to the robustness of the models without affecting it's accuracy.

2. In the imitation learning model the collision rate decreased by 68.9% on average in our reproduced results with the time taken showing little deviation (Table 6).

3. The Social-NCE addition to the Rainbow-DQN based reinforcement learning model, as in the original paper, achieves a reward of 0.6 in 2000 episodes in comparison to 4000 episodes of the original Reinforcement Learning model(Table 7).

The hyperparameter tuning conducted was also vastly helpful and lead to an increase of accuracy by 0.91 %. The loss hyperparameters, determined the sensitivity of the model. The contrastive weight, determined emphasis of the Social-NCE loss. The more the emphasis, the better the model learnt to differentiate between a positive and negative sample, but at the expense of loss of proximity to the actual training examples. It remains difficult to analytically understand the effect of change in the temperature hyperparameter.

Hyperparameter search, even though tedious, can lead to a great increase in accuracy. The tuning of hyperparameters involved in the model, lead to an overall increase in accuracy. In task like trajectory prediction and motion forecasting, it might be crucial to try and increase the accuracy as much as possible. However, one thing to be noted is that the Social-STGCNN had a huge running time, and one sweep took a huge amount of time.

The effect of the Data Augmentation hyperparameters, seem to be highly variable. It is natural that results are likely to vary greatly with the choice of dataset, and the nature of the problem statement. This is because these hyperparameters are physical constraints put on the model, and hence might lead to different results for different datasets.

Further, it was found that best results were found when the value of the contrastive weight to be 16, while the default value was 2. The values might differ distinctly, but this reinforces confidence in the proposed Social-NCE loss.

### 5.1 What was easy

The authors have provided a detailed public codebase on the implementation of Social-NCE on Trajectron++, Social-STGCNN and imitation learning model. Further, they shared the codebase for the reinforcement learning model. All the codebases have instructions on how to set up the environment and logs the important metrics, which proved to be helpful in reproduction.

### 5.2 What was difficult

The porting of the implementation of Social-NCE in the Trajectron++ and Social-STGCNN models from PyTorch to PyTorch Lightning required an understanding of those models and their original codebase which required additional time. Training of the models from scratch required large computation power. All the training was done over cloud GPUs with limited runtimes which often fell short of the time required for training.

### 5.3 Communication with original authors

We mailed the authors listing down some of the queries we had on their code implementation. We also had some queries regarding the important hyperparameters that we could tune to improve model performance. The authors gave a prompt reply to our questions. They shared the codebase for the reinforcement learning model as well. Their contribution has helped us with some crucial points in the report.

## 6 Future Work

We originally planned to perform the following additional experiments which we couldn't finish due to lack of time. They have been listed down below and we believe that future work on this paper can be done in this direction.

- A best hyperparameter search on Social-NCE in Trajectron++, the imitation learning model and the Rainbow-DQN based Reinforcement Learning model as well and comparison of the variation in results for different models.

- Implementation of Social-NCE on the Social-LSTM and Directional-LSTM models on the Trajnet++ benchmark, the results for which have been given in the original paper.

- Implementation of Social-NCE on state of the art models in other benchmarks such as on the PGP model [9] for the nuScences dataset.

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
