# OpenReview forum: "Reproducibility Report: Contrastive Learning of Socially-aware Motion Representations"
_ML_Reproducibility_Challenge/2021/Fall — RC2021_

### Official Review · Reviewer_xuqF · 2022-03-01
**Interesting report, a implications of hyper parameters?**

**Rating:** 6
**Confidence:** 5

**Review:**

One of the more interesting characteristics of this reproducibility paper is the detailed information about the loss functions under study as well as the details of hyper parameter tuning.
With respect to the formatting, the first page is present and the template stylesheet was used for the paper.  The paper satisfactorily states the scope of reproducibility, and adheres to it. Although, most results claimed in the paper are verified, the Social-LSTM and Directional-LSTM models were not due to lack of time.   With respect to whether reproduced from scratch or re-used author repository: The hyperparameter search is shown in Section 3.3 . No additional codebase was attached with readable code/docs because they used the original authors code.
The authors did communicate with the original authors who also provided code and data used during the replication study.  The hyperparameter search is shown in Section 3.3. It could have been interesting to read your insights you obtained from the hyper parameter analysis in addition to the details you provided. The ablation study is mentioned in the paper but not detailed in it.   The report provides a detailed discussion of the results and the reproducibility. The report also details the difficult and easy aspects to reproduce from the original paper, namely, the good quality of the code and the contrastive loss implementation, respectively.   The authors of the report do not directly address the original authors in their recommendations but their general recommendations and the queries to the authors could be used by the original authors to improve the reproducibility of the paper.  Finally, this paper is well organized and easy to follow.

---

### Official Review · Reviewer_QyXq · 2022-03-07
**review for submission# 83**

**Rating:** 7
**Confidence:** 4

**Review:**

The authors did a good job in reproducing the algorithm presented in "Social NCE: Contrastive Learning of Socially-aware Motion Representations" published in ICCV 2021, and validated the claims in the paper.

1. The scope of the report is clearly defined.
2. The authors compared both the repository given by the original authors as well as a re-implementation of the algorithm by their own, which is very valuable to assess the reproducibility of the original work. The re-implemented version was also used to run additional results that weren't presented in the original work.
3. The authors did good hyper-parameter search and communicated with the original authors to unblock challenges in the experiments to make sure a fair comparison is done.

---

### Official Review · Reviewer_26LG · 2022-03-07
**ML Reproducibility Challenge 2021 - Contrastive Learning of Socially-aware Motion Representations**

**Rating:** 9
**Confidence:** 4

**Review:**

The authors tried to reproduce the work conducted in the paper 'Contrastive Learning of Socially-aware
Motion Representations'. They were able to successfully reproduce the paper with a slight deviation of 2% in their results. The analysis, discussion and the results presented by the authors here is very thorough. They also have had discussions with the original authors of the paper to get clarity on certain components. The format of the reproducibility report is also very good and easy to understand and comprehend. The authors were not able to reproduce all the results in the given time however.

---

### Meta-Review · Area_Chair_jaLD · 2022-04-08

**Recommendation:** Accept
**Confidence:** 5

**Metareview:**

Good report, well written and hyperparameter search conducted fairly. However, the paper lacks on analysis and ablation studies, which could have strengthened the paper, which makes the paper borderline. However, reviewers agree that the report is well written, easy to follow and detailed discussion on the state of reproducibility. Thus, I recommend the paper for acceptance in the venue.

---

### Decision · Program_Chairs · 2022-04-09

**Decision:**

Accept

**Comment:**

Following the recommendation of reviewers and meta-reviewer, the paper is accepted for ML Reproducibility Challenge 2021, and will be published in the upcoming special edition of ReScience Journal.